



# Importance of interpolation and coincidence errors in data fusion

Simone Ceccherini [1], Bruno Carli [1], Cecilia Tirelli [1], Nicola Zoppetti [1], Samuele Del Bianco [1], Ugo Cortesi [1], Jukka Kujanpää [2], Rossana Dragani [3]

[1]Istituto di Fisica Applicata "Nello Carrara" del Consiglio Nazionale delle Ricerche, Via Madonna del Piano 10, 50019 Sesto Fiorentino, Italy
[2]Finnish Meteorological Institute, Earth Observation Unit, P.O. Box 503, FI-00101 Helsinki, Finland
[3]European Centre for Medium-Range Weather Forecasts, Shinfield Park, Reading, RG2 9AX, United Kingdom

*Correspondence to*: Simone Ceccherini (S.Ceccherini@ifac.cnr.it)

**Abstract.** The Complete Data Fusion method is applied to ozone profiles obtained from simulated measurements in the ultraviolet and in the thermal infrared in the framework of the Sentinel 4 mission of the Copernicus programme. We observe that the quality of the fused products is degraded when the fusing profiles are either retrieved on different vertical grids or referred to different true profiles. To address this shortcoming, a generalization of the complete data fusion method, which takes into account interpolation and coincidence errors, is presented. This upgrade overcomes the encountered problems and provides products of good quality when the fusing profiles are both retrieved on different vertical grids and referred to different true profiles. The impact of the interpolation and coincidence errors on number of degrees of freedom and errors of the fused profile is also analyzed. The approach developed here to account for the interpolation and coincidence errors can also be followed to include other error components, such as forward model errors.

## 1 Introduction

Many remote sensing observations of vertical profiles of atmospheric variables are obtained with instruments operating on space-borne and airborne platforms, as well as from ground-based stations. Recently, the Complete Data Fusion (CDF) method (Ceccherini et al., 2015) was proposed to combine independent measurements of the same profile in order to exploit all the available information for a comprehensive and concise description of the atmospheric state. This is an a posteriori method that uses standard retrieval products. With simple implementation requirements, the CDF products are equivalent to those from a simultaneous retrieval, considered to be the most comprehensive way of exploiting different observations of the same quantity (Aires et al., 2012), in spite of a greater computational complexity. However, so far, the data fusion method was mainly applied to measurements performed by the same instrument while sounding the same air sample.

Limited tests were conducted on measurements performed by different instruments when inconsistencies due to differences in the observed true profiles (because of the non-perfect coincidence of the space-time location of the measurements) could degrade the optimal performances of the simultaneous retrieval. About the fusion of data provided by different instruments, it has been proved (Ceccherini, 2016) that the CDF is completely equivalent to the measurement space solution (MSS) data fusion method (Ceccherini et al., 2009). The latter was successfully applied to the data fusion of MIPAS-ENVISAT and IASI-METOP measurements (Ceccherini et al, 2010a; Ceccherini et al, 2010b) and of MIPAS-STR and MARSCHALS measurements (Cortesi et al., 2016). However, as in these cases the measurements to be fused (referred to as fusing profiles hereafter) carried information about basically complementary altitude ranges, their possible inconsistency did not result in unrealistic fused profiles.

The first applications of data fusion were made with profiles retrieved on the same vertical grid. A first analysis of the effect of different grids on the quality of the fused products was performed and presented by Ceccherini et al. (2016). In this case, the individual profiles were first obtained on grids optimally defined according to the information content of the individual observations. Then, the CDF was performed using averaging kernel matrices (AKMs) interpolated to a common grid optimized for the data fusion product. Compared to the case in which the individual retrievals are obtained directly on the



grid optimized for the data fusion, the number of degrees of freedom (DOF) is with this approach reduced of about a quarter. Thus, in data fusion applications the choice of the retrieval grid can lead to an information content loss that cannot be restored with interpolation.

Here, we consider the general problem posed by the application of the CDF to measurements performed by different instruments that are retrieved on different vertical grids and referring to different true profiles. We analyze this problem using simulated measurements of ozone profiles obtained in the ultraviolet and in the thermal infrared in the framework of the Sentinel 4 (S4) mission (ESA, 2017) of the Copernicus programme (http://www.copernicus.eu/main/sentinels). The advantages to use a multispectral approach for observing the ozone profile from space by synergism of atmospheric radiances in the thermal infrared and in the ultraviolet has been studied by Landgraf and Hasekamp (2007), Worden et al.

(2007), Fu et al. (2013), Natraj et al., (2011), Cuesta et al. (2013) and Costantino et al. (2017).

The paper is organized as follows: Section 2 presents an account of the problems that occur when the CDF is applied to vertical profiles retrieved on different vertical grids and referring to different true profiles. In Section 3, we theoretically analyze the problems discussed in Section 2, and show how the CDF method can be modified to overcome them. In Section 4, we show how the solution proposed in Section 3 solves the problems discussed in Section 2. In Section 5, we describe

how to deal with forward model errors. Conclusions are drawn in Section 6.

## 2 Application of the CDF to profiles retrieved on different vertical grids and related to different true profiles

The future atmospheric Sentinel missions of the Copernicus programme (http://www.copernicus.eu/main/sentinels) will provide great scope and a real testbed for data fusion applications. The wealth of data that will become available from these missions will likely present technical challenges to many applications. With the use of data fusion, the number of products

can be reduced while maintaining the information content of the original datasets. For this reason, we test the CDF method on simulated data of the S4. We simulate two S4 ozone vertical profile measurements as they could be obtained from the Infrared Sounder (IRS) in the thermal infrared and from the Ultraviolet, Visible and Near-Infrared Sounding (UVN) spectrometer in the ultraviolet (http://www.eumetsat.int/website/home/Satellites/FutureSatellites/MeteosatThirdGeneration/MTGDesign/index.html)

onboard the MTG (Meteosat Third Generation) satellite. We refer to these two simulated measurements as TIR measurement and UV measurement, respectively.

In order to evaluate the effect of the variability of vertical grids and of true profiles, we have analyzed three cases when:

1.    The simulated measurements refer to the same true profile and are retrieved on the same vertical grid.
2.    The simulated measurements refer to the same true profile but are retrieved on different vertical grids.

3.    The simulated measurements refer to different true profiles and are retrieved on the same vertical grid.

In all three cases, the true profile and the vertical grid of the UV measurement are kept fixed and, when pertinent, are changed for the TIR measurement. For simplicity, we define the vertical grid of the data fusion product to coincide with the fixed grid of the UV measurement. In the following, the vertical grid of the fusion product is referred to as the fusion grid.

For a meaningful comparison of the quality of fusing and fused profiles, it is necessary to have common a priori profiles and

common a priori covariance matrices (CMs). Therefore, the a priori of the fusing profiles, which are produced with individual a priori assumptions, have been modified using the method described in (Ceccherini et al., 2014). In the comparisons, we use the same a priori profiles provided by the McPeters and Labow climatology (McPeters and Labow, 2012) for all fusing and fused profiles. The a priori CMs are obtained using the standard deviation of the McPeters and Labow climatology when its value is larger than 20% of the a priori profile and a value of 20% of the a priori profile in the

other cases. The off diagonal elements are calculated considering a correlation length of 6 km.





The results obtained in the three test cases are reported in Figures 1-3. These figures show in the left panel the true profiles, in the central panel the mean value of the true profiles and the profiles obtained from the measurements (TIR, UV and data fusion) and in the right panel the residuals, i.e. the differences between the three estimated profiles and the mean value of the true profiles.

We observe that, while in case 1 the differences between the profile obtained from the fusion and the mean of the true profiles are smaller than, or comparable to, those of the profiles obtained from the TIR and UV measurements, in cases 2 and 3 these differences are significantly larger. Therefore, in cases 2 and 3 the fusion provides a product of poorer quality than that of the single products.

These tests show that the CDF algorithm and the equivalent simultaneous retrieval work well in case 1, while they have
problems in cases 2 and 3, where the profiles are retrieved on different vertical grids and are referred to different true profiles, respectively.

The problem encountered in case 2 is due to the fact that the data fusion is made using estimates of the AKMs on the fusion grid (see Subsection 3.1) obtained by interpolation of the original AKMs (Ceccherini et al., 2016), which are only an approximation of the real AKMs on the fusion grid. We refer to this effect as *interpolation error*. The problem encountered
in case 3 is related to different true profiles and we refer to this effect as *coincidence error* because it occurs when fusing profiles that do not correspond to the same space-time location.

## 3 Method

In this section, we report a theoretical analysis performed to overcome the problems highlighted in the previous section. In Subsection 3.1, we recall the formulas of the CDF method in order to establish the formalism subsequently used in
Subsection 3.2, where an upgrade of the method is proposed.

### 3.1 CDF

We assume to have $N$ independent simultaneous measurements of the vertical profile of an atmospheric target referred to the same space-time location. Performing the retrieval of the $N$ measurements with the optimal estimation method (Rodgers, 2000), we obtain $N$ vectors $\hat{\mathbf{x}}_i$ ($i$=1, 2, …, $N$) that provide independent estimates of the profile here assumed to be
represented on a common vertical grid. The use of a priori information ensures the possibility of having a common retrieval grid also in the case of observations with different vertical coverage.

The vectors $\hat{\mathbf{x}}_i$ are characterized by the CMs $\mathbf{S}_i$ and the AKMs $\mathbf{A}_i$ (Ceccherini et al., 2003; Ceccherini and Ridolfi, 2010; Rodgers, 2000):

$$\mathbf{S}_i \equiv \left\langle \boldsymbol{\sigma}_i \boldsymbol{\sigma}_i^T \right\rangle = \left( \mathbf{K}_i^T \mathbf{S}_{yi}^{-1} \mathbf{K}_i + \mathbf{S}_{ai}^{-1} \right)^{-1} \mathbf{K}_i^T \mathbf{S}_{yi}^{-1} \mathbf{K}_i \left( \mathbf{K}_i^T \mathbf{S}_{yi}^{-1} \mathbf{K}_i + \mathbf{S}_{ai}^{-1} \right)^{-1}, \tag{1}$$

$$\mathbf{A}_i \equiv \frac{\partial \hat{\mathbf{x}}_i}{\partial \mathbf{x}} = \left( \mathbf{K}_i^T \mathbf{S}_{yi}^{-1} \mathbf{K}_i + \mathbf{S}_{ai}^{-1} \right)^{-1} \mathbf{K}_i^T \mathbf{S}_{yi}^{-1} \mathbf{K}_i, \tag{2}$$

where $\boldsymbol{\sigma}_i$ are the errors on $\hat{\mathbf{x}}_i$ obtained propagating the errors of the observations through the retrieval processes (noise
errors), $\mathbf{K}_i$ are the Jacobians of the forward models, $\mathbf{S}_{yi}$ are the CMs of the observations, $\mathbf{S}_{ai}$ are the CMs of the a priori profiles and $\mathbf{x}$ is the true profile.

The CDF of the considered profiles is given by (see Ceccherini et al., 2015)

$$\mathbf{x}_f = \left( \sum_{i=1}^{N} \mathbf{A}_i^T \mathbf{S}_i^{-1} \mathbf{A}_i + \mathbf{S}_a^{-1} \right)^{-1} \left( \sum_{i=1}^{N} \mathbf{A}_i^T \mathbf{S}_i^{-1} \boldsymbol{\alpha}_i + \mathbf{S}_a^{-1} \mathbf{x}_a \right), \tag{3}$$

where



$$\boldsymbol{\alpha}_i \equiv \hat{\mathbf{x}}_i - \left(\mathbf{I} - \mathbf{A}_i\right)\mathbf{x}_{ai} = \mathbf{A}_i\mathbf{x} + \boldsymbol{\sigma}_i , \tag{4}$$

$\mathbf{x}_{ai}$ is the a priori profile used in the $i$-th retrieval, $\mathbf{x}_a$ and $\mathbf{S}_a$ are the a priori profile and its CM used to constrain the data fusion.

We note that the vector $\boldsymbol{\alpha}_i$, which can be calculated from the available retrieval products, is a measurement of the vector $\mathbf{x}$, made using the rows of the AKM $\mathbf{A}_i$, and it has the same errors $\boldsymbol{\sigma}_i$ as the retrieved profile $\hat{\mathbf{x}}_i$, therefore, it is characterized by the CM $\mathbf{S}_i$.

The fused profile has a CM, obtained propagating the errors of $\boldsymbol{\alpha}_i$ into $\mathbf{x}_f$, equal to

$$\mathbf{S}_f = \left(\sum_{i=1}^{N}\mathbf{A}_i^T\mathbf{S}_i^{-1}\mathbf{A}_i + \mathbf{S}_a^{-1}\right)^{-1}\sum_{i=1}^{N}\mathbf{A}_i^T\mathbf{S}_i^{-1}\mathbf{A}_i\left(\sum_{i=1}^{N}\mathbf{A}_i^T\mathbf{S}_i^{-1}\mathbf{A}_i + \mathbf{S}_a^{-1}\right)^{-1} \tag{5}$$

and an AKM, obtained performing the derivative of $\mathbf{x}_f$ with respect to the true profile, equal to

$$\mathbf{A}_f = \left(\sum_{i=1}^{N}\mathbf{A}_i^T\mathbf{S}_i^{-1}\mathbf{A}_i + \mathbf{S}_a^{-1}\right)^{-1}\sum_{i=1}^{N}\mathbf{A}_i^T\mathbf{S}_i^{-1}\mathbf{A}_i . \tag{6}$$

The CDF formula (Eq. (3)) involves a summation of AKMs made possible by the common grid. When the fusing profiles $\hat{\mathbf{x}}_i$ are represented on different vertical grids, the available AKMs are also defined on different vertical grids, thus in this case, it is necessary to perform a resampling of the AKMs (Calisesi et al., 2005), which makes their second index equal to that of the common fusion grid. Following Ceccherini et al. (2016), we define such a transformation as follows:

$$\mathbf{A}_i{}' = \mathbf{A}_i\mathbf{R}_i , \tag{7}$$

where $\mathbf{R}_i$ are the generalized inverse matrices of the linear interpolation matrices $\mathbf{H}_i$, which interpolate the profiles on the fusing grids to the fusion grid. In this case, using Eq. (7), Eq. (3) becomes:

$$\mathbf{x}_f = \left(\sum_{i=1}^{N}\mathbf{R}_i^T\mathbf{A}_i^T\mathbf{S}_i^{-1}\mathbf{A}_i\mathbf{R}_i + \mathbf{S}_a^{-1}\right)^{-1}\left(\sum_{i=1}^{N}\mathbf{R}_i^T\mathbf{A}_i^T\mathbf{S}_i^{-1}\boldsymbol{\alpha}_i + \mathbf{S}_a^{-1}\mathbf{x}_a\right). \tag{8}$$

### 3.2 Interpolation and coincidence errors

Let us first consider the interpolation error. The vectors $\boldsymbol{\alpha}_i$, defined by Eq. (4), are measurements of the true profile, each made with the averaging kernels $\mathbf{A}_i$. Let us assume that each measurement is defined on a different retrieval grid, identified by the same index that identifies the measurements, then Eq. (4) becomes:

$$\boldsymbol{\alpha}_i = \mathbf{A}_i\mathbf{x}_i^{(i)} + \boldsymbol{\sigma}_i , \tag{9}$$

where $\mathbf{x}_i^{(i)}$ is the true profile related to the $i$-th measurement that, by definition, is sampled with the $i$-th grid, as highlighted by the superscript in parenthesis.

Eq. (8) shows that in presence of different vertical grids the CDF combines measurements with sensitivity to the true profile expressed by $\mathbf{A}_i\mathbf{R}_i$. This operation assumes that we are combining measurements on the common fusion grid, i.e. measurements of $\mathbf{A}_i\mathbf{R}_i\mathbf{x}_i^{(f)}$, with $\mathbf{x}_i^{(f)}$ being the true profile related to the $i$-th measurement represented on the fusion grid. If using $\boldsymbol{\alpha}_i$ (Eq. (9)), which is the measurement of $\mathbf{A}_i\mathbf{x}_i^{(i)}$, the estimate of the required measurement $\mathbf{A}_i\mathbf{R}_i\mathbf{x}_i^{(f)}$ is made with an error equal to $\mathbf{A}_i\mathbf{x}_i^{(i)} - \mathbf{A}_i\mathbf{R}_i\mathbf{x}_i^{(f)}$.

We can explicitly introduce this error in the expression of $\boldsymbol{\alpha}_i$ by rearranging Eq. (9) in the following way:

$$\boldsymbol{\alpha}_i = \mathbf{A}_i\mathbf{R}_i\mathbf{x}_i^{(f)} + \mathbf{A}_i\left(\mathbf{x}_i^{(i)} - \mathbf{R}_i\mathbf{x}_i^{(f)}\right) + \boldsymbol{\sigma}_i . \tag{10}$$

It is useful to introduce for $\mathbf{x}_i^{(i)}$ and $\mathbf{x}_i^{(f)}$ the following notations:



$$\mathbf{x}_i^{(i)} = \mathbf{C}^{(i)}\mathbf{x}_i, \tag{11}$$

$$\mathbf{x}_i^{(f)} = \mathbf{C}^{(f)}\mathbf{x}_i, \tag{12}$$

where $\mathbf{x}_i$ is the true profile related to the $i$-th measurement represented on a very fine grid that includes all the levels of the fusion grid ($f$) and of the N grids ($i$). $\mathbf{C}^{(i)}$ and $\mathbf{C}^{(f)}$ are the sampling matrices from the fine grid to the grids ($i$) and to the grid ($f$), respectively.

Substituting Eqs (11) and (12) in Eq. (10), we obtain

$$\boldsymbol{\alpha}_i = \mathbf{A}_i\mathbf{R}_i\mathbf{C}^{(f)}\mathbf{x}_i + \mathbf{A}_i\left(\mathbf{C}^{(i)} - \mathbf{R}_i\mathbf{C}^{(f)}\right)\mathbf{x}_i + \boldsymbol{\sigma}_i. \tag{13}$$

Let us now also consider the coincidence error. In general, we fuse measurements made in different space-time locations within a given coincidence criterion. These measurements correspond to different true profiles and the purpose of the data fusion can be the determination of either the mean value of these true profiles or the true profile in a given space-time location identified as the central point of the coincidence intervals. We indicate with $\overline{\mathbf{x}}$ the unknown profile estimated by the data fusion. If we introduce the quantity $\boldsymbol{\sigma}_{i,coin}$, which gives the deviation of $\mathbf{x}_i$ from the unknown profile $\overline{\mathbf{x}}$ :

$$\mathbf{x}_i = \overline{\mathbf{x}} + \boldsymbol{\sigma}_{i,coin}, \tag{14}$$

Eq. (13) becomes:

$$\begin{aligned}
\boldsymbol{\alpha}_i &= \mathbf{A}_i\mathbf{R}_i\mathbf{C}^{(f)}\overline{\mathbf{x}} + \mathbf{A}_i\left(\mathbf{C}^{(i)} - \mathbf{R}_i\mathbf{C}^{(f)}\right)\overline{\mathbf{x}} + \mathbf{A}_i\mathbf{C}^{(i)}\boldsymbol{\sigma}_{i,coin} + \boldsymbol{\sigma}_i = \\
&= \mathbf{A}_i\mathbf{R}_i\overline{\mathbf{x}}^{(f)} + \mathbf{A}_i\left(\mathbf{C}^{(i)} - \mathbf{R}_i\mathbf{C}^{(f)}\right)\overline{\mathbf{x}} + \mathbf{A}_i\mathbf{C}^{(i)}\boldsymbol{\sigma}_{i,coin} + \boldsymbol{\sigma}_i.
\end{aligned} \tag{15}$$

after using Eq. (12) for $\overline{\mathbf{x}}$ .

An estimate of the quantity $\mathbf{A}_i\left(\mathbf{C}^{(i)} - \mathbf{R}_i\mathbf{C}^{(f)}\right)\overline{\mathbf{x}}$ can be obtained writing $\overline{\mathbf{x}}$ as the a priori profile plus the deviation $\boldsymbol{\sigma}_a$ from it:

$$\overline{\mathbf{x}} = \mathbf{x}_a + \boldsymbol{\sigma}_a. \tag{16}$$

Substituting Eq. (16) in Eq. (15) and rearranging the terms of the equation, we can define a new quantity, $\tilde{\boldsymbol{\alpha}}_i$ , equal to

$$\begin{aligned}
\tilde{\boldsymbol{\alpha}}_i &\equiv \boldsymbol{\alpha}_i - \mathbf{A}_i\left(\mathbf{C}^{(i)} - \mathbf{R}_i\mathbf{C}^{(f)}\right)\mathbf{x}_a = \\
&= \mathbf{A}_i\mathbf{R}_i\overline{\mathbf{x}}^{(f)} + \mathbf{A}_i\left(\mathbf{C}^{(i)} - \mathbf{R}_i\mathbf{C}^{(f)}\right)\boldsymbol{\sigma}_a + \mathbf{A}_i\mathbf{C}^{(i)}\boldsymbol{\sigma}_{i,coin} + \boldsymbol{\sigma}_i.
\end{aligned} \tag{17}$$

Each $\tilde{\boldsymbol{\alpha}}_i$ is a measurement of $\overline{\mathbf{x}}^{(f)}$ made using the rows of the matrix $\mathbf{A}_i\mathbf{R}_i$ and a total error given by the sum of the noise error $\boldsymbol{\sigma}_i$ plus the terms $\mathbf{A}_i\left(\mathbf{C}^{(i)} - \mathbf{R}_i\mathbf{C}^{(f)}\right)\boldsymbol{\sigma}_a$ and $\mathbf{A}_i\mathbf{C}^{(i)}\boldsymbol{\sigma}_{i,coin}$ that can be interpreted as the interpolation error and the coincidence error, respectively.

For the estimate of the interpolation error, we use the a priori CM $\mathbf{S}_a$ of $\boldsymbol{\sigma}_a$ and, therefore, the interpolation error is characterized by the CM:

$$\mathbf{S}_{i,\mathrm{int}} = \mathbf{A}_i\left(\mathbf{C}^{(i)} - \mathbf{R}_i\mathbf{C}^{(f)}\right)\mathbf{S}_a\left(\mathbf{C}^{(i)} - \mathbf{R}_i\mathbf{C}^{(f)}\right)^T\mathbf{A}_i^T. \tag{18}$$

To characterize the coincidence error, we introduce the CM $\mathbf{S}_{coin}$ of $\boldsymbol{\sigma}_{i,coin}$. If $\overline{\mathbf{x}}$ represents the mean value of the true profiles, $\mathbf{S}_{coin}$ accounts for the dispersion of the true profiles, thus it depends on the coincidence criteria and it is the same for all the measurements to be fused together. If $\overline{\mathbf{x}}$ represents the true profile in a specific space-time location, $\mathbf{S}_{coin}$ is zero if the measurement is exactly in that location and it increases going away from that location. The values of $\mathbf{S}_{coin}$ as a function of





space-time location should reflect the variability of the true profile with the location. Then, the coincidence error is characterized by the CM

$$\mathbf{S}_{i,coin} = \mathbf{A}_i \mathbf{C}^{(i)} \mathbf{S}_{coin} \mathbf{C}^{(i)T} \mathbf{A}_i^{\,T}.$$ (19)

In conclusion, the CDF formula, given by Eq. (3), can be modified to account for the interpolation and coincidence errors by replacing $\boldsymbol{\alpha}_i$ with

$$\tilde{\boldsymbol{\alpha}}_i = \boldsymbol{\alpha}_i - \mathbf{A}_i \left( \mathbf{C}^{(i)} - \mathbf{R}_i \mathbf{C}^{(f)} \right) \mathbf{x}_a$$ (20)

and $\mathbf{S}_i$ with

$$\tilde{\mathbf{S}}_i = \mathbf{S}_i + \mathbf{S}_{i,\text{int}} + \mathbf{S}_{i,coin}.$$ (21)

The CM given by Eq. (21) is also used in place of $\mathbf{S}_i$ in Eqs (5, 6) for the calculation of the CM and AKM of the fused profile.

**4 Tests with the upgraded algorithm: results and discussion**

**4.1 The effect on fused profiles**

We repeated the test cases of fusion 2 and 3 shown in Section 2 with the modified method described in Subsection 3.2.

In Figures 4 and 5, we report the noise errors, the interpolation errors and the coincidence errors related, respectively, to case 2 and case 3, for both TIR and UV measurements. These errors are calculated as the square root of the diagonal elements of $\mathbf{S}_i$, $\mathbf{S}_{i,int}$ and $\mathbf{S}_{i,coin}$, respectively. In case 2, the vertical grids are different for the two measurements and since the fusion grid coincides with the vertical grid of the UV measurement, the interpolation errors are different from zero for the TIR

measurement and equal to zero for the UV measurement. The coincidence errors are equal to zero in both TIR and UV measurements because the true profiles are the same. In case 3, the interpolation errors are equal to zero for both TIR and UV measurements because the fusion grid coincides with that of the fusing profiles. The coincidence errors are instead different from zero because the true profiles are different and their CMs, chosen equal for both TIR and UV measurements, are obtained considering an error of 5% of the a priori profile and a correlation length of 6 km.

Figures 6 and 7 show the same quantities reported in central and right panels of Figures 2 and 3, respectively. In these, we have added the fused profiles and the residuals obtained with the modified algorithm. We can see that, in both tests, the modified method provides residuals that are significantly smaller than those obtained with the original CDF method.

These tests show that the upgrade of the CDF method proposed in Subsection 3.2 solves the problems observed in Section 2 that occur when either the fusing profiles are retrieved on different vertical grids or they refer to different true profiles. The

modified method is a generalization of the CDF that allows its application to a wide-range of cases.

**4.2 The effect on errors and number of DOF**

We now look at the effect of the generalized method on the errors and on the number of DOF. Figures 8 and 9 show the errors of the fused profile when we use either the original or the modified method for the cases 2 and 3, respectively. These errors are calculated as the square root of the diagonal elements of $\mathbf{S}_f$ given in Eq. (5), where, in the modified method, $\mathbf{S}_i$ is

replaced by $\tilde{\mathbf{S}}_i$. For the three cases described in Section 2, Table 1 gives the number of DOF of the profiles obtained from the individual TIR and UV measurements, and from the CDF using both the original and the generalized formulation. The numbers of DOF are calculated as the trace of the AKMs. For the fused products the AKM is $\mathbf{A}_f$ given by Eq. (6), where, in the generalized formulation, $\mathbf{S}_i$ is replaced by $\tilde{\mathbf{S}}_i$.



The introduction of the interpolation error (case 2) does not significantly modify the errors and determines a decrease of the number of DOF of the fused profile of about 1. The introduction of the coincidence error (case 3) determines a significant increase of the errors and a small decrease of the number of DOF of the fused profile equal to about 0.5. However, in both cases the number of DOF of the fused profile obtained with the modified method is larger than the number of DOF of the individual fusing profiles, proving the information gain provided by the fusion.

From the analysis of errors and number of DOF we deduce that the interpolation error has the largest impact on the vertical resolution, while the coincidence error has the largest impact on the errors. However, these numerical results depend on the values that interpolation and coincidence errors have in the single cases.

## 5 Other error sources

In this paper, we considered simulated measurements, which generally do not include all the error components that are present in real measurements. When real measurements are considered, there are other important error sources that can cause inconsistency among the fusing profiles, such as forward model errors, due for example to approximations in the model and uncertainties in atmospheric and instrumental parameters. When performing data fusion, these errors can also lead to quality loss and show problems similar to those described in Section 2. These problems can be avoided by accounting for them in the CDF formulation. In particular, Eq. (21) can be modified to account for an extra CM term, $\mathbf{S}_{i,other}$, as follows:

$$\tilde{\mathbf{S}}_i = \mathbf{S}_i + \mathbf{S}_{i,other} + \mathbf{S}_{i,\text{int}} + \mathbf{S}_{i,coin}. \tag{22}$$

## 6 Conclusions

We analyzed the problem posed by the application of the CDF to vertical profiles obtained with different instruments, which use different retrieval grids and observe different true profiles. To this purpose, we studied simulated ozone profile measurements expected from the MTG payload for the S4 mission of the Copernicus programme: namely, those provided by the IRS in the thermal infrared and by the UVN spectrometer in the ultraviolet. The study showed that the CDF algorithm works well when the fusing profiles are represented on the same vertical grid and refer to the same true profile, otherwise, if these identities are not present, the algorithm provides unsatisfactory results because the fused profile differs from the mean of the true profiles significantly more than the fusing profiles. Indeed, the CDF, which exploits all the existing information for the determination of the best fused profile, is misled by the inconsistent information and provides unrealistic fused profiles.

In order to overcome this new problem, we performed a theoretical analysis that led to a generalization of the CDF method to the cases in which interpolation and coincidence errors occur. The interpolation error is present when the vertical grids of the fusing profiles differ from the fusion grid, meaning that an interpolation of the AKMs is necessary. In this case, the interpolated AKMs are only an approximation of the real AKMs on the fusion grid. The coincidence error is a consequence of the fact that the fusing profiles are not generally co-located in space and time, thus referring to different true profiles.

The generalized algorithm allows for these inconsistencies and provides fused profiles that are in better agreement with the true profiles than those obtained with the original CDF algorithm.

With the algorithm generalization, the fusion provides in general fused profiles that are also better than the fusing profiles in terms of total error and number of DOF. However, a more comprehensive error budget, which may even cause the fused profile to have larger errors than the fusing profiles (indeed coincidence and interpolation errors do not have to be considered for the individual fusing profiles), is now considered. If neither of the qualifiers (total error and number of DOF) is improved, the fusion process is not justified.



An approach similar to that used to account for interpolation and coincidence errors can also be useful to include other error components, such as forward model errors, in the fusion process.

*Author contribution*. SC deduced the expression of the interpolation and coincidence errors and wrote the draft version of the paper. BC suggested the idea to introduce the interpolation and coincidence errors and contributed to the interpretation of the results. NZ wrote the Python code of the complete data fusion. CT and SDB performed the simulation of the infrared measurements. JK performed the simulation of the ultraviolet measurements. UC put together the team of authors and coordinated its activity. RD deeply revised the manuscript.

*Data availability*. The data of the simulations presented in the paper are available upon request to the authors.

*Competing interests*. The authors declare that they have no conflict of interest.

*Acknowledgments*. The results presented in this paper arise from research activities conducted in the framework of the AURORA project (http://www.aurora-copernicus.eu/) supported by the Horizon 2020 research and innovation programme of
the European Union (Call: H2020-EO-2015; Topic: EO-2-2015) under Grant Agreement N. 687428.

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



**Table 1. Number of DOF of the profiles obtained with the TIR measurement, the UV measurement, the original fusion method and the modified fusion method for each of the three cases described in Section 2.**

|        | TIR | UV  | FUS | FUS new |
|--------|-----|-----|-----|---------|
| Case 1 | 4.7 | 3.2 | 5.6 |         |
| Case 2 | 4.7 | 3.2 | 5.8 | 4.9     |
| Case 3 | 4.8 | 3.2 | 5.6 | 5.1     |



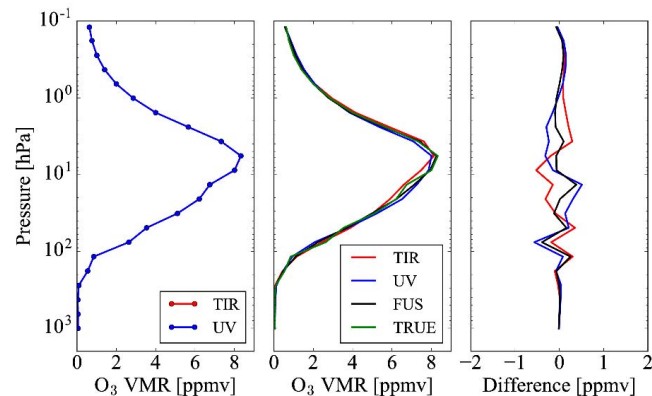

**Figure 1.** Left panel: true ozone profiles related to TIR (red line) and UV (blue line) measurements. Central panel: ozone profiles obtained from TIR measurement (red line), from UV measurement (blue line), from the data fusion (black line) compared with the
10  mean value of the true profiles (green line). Right panel: residual errors obtained as differences of the ozone profiles obtained from TIR measurement (red line), from UV measurement (blue line) and from data fusion (black line) from the mean value of true profiles. All the reported quantities are related to case 1.





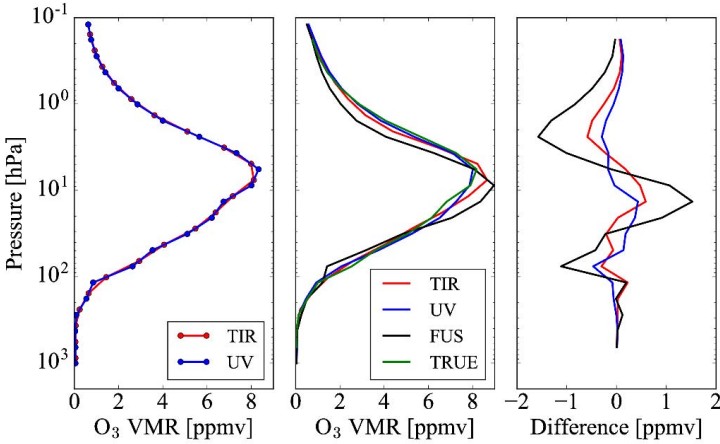

**Figure 2. As Figure 1 in case 2.**





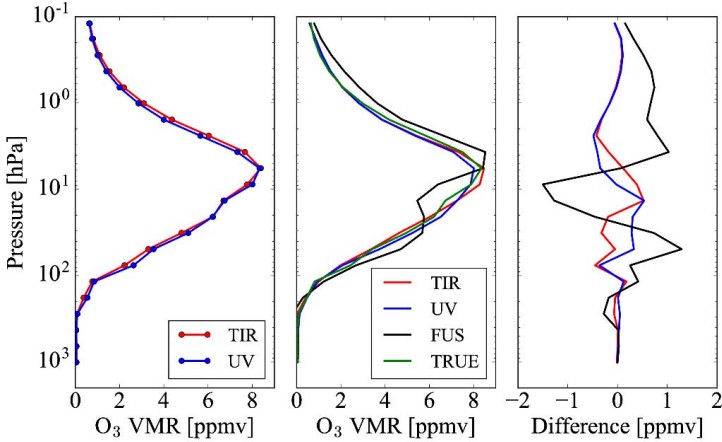

**Figure 3. As Figure 1 in case 3.**





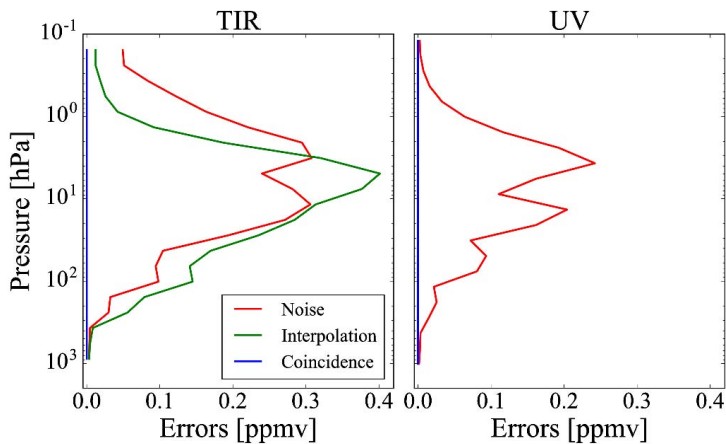

**Figure 4. Noise errors (red lines), interpolation errors (green lines) and coincidence errors (blue lines) in case 2 for TIR and UV**
10 **measurements.**



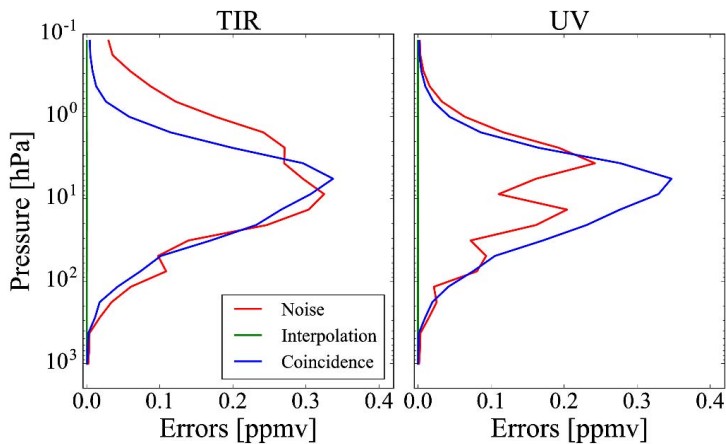

**Figure 5. As Fig. 4 in case 3.**





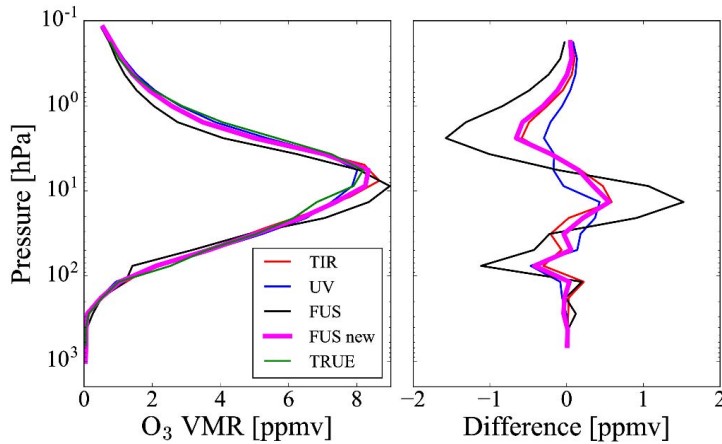

**Figure 6. Central and right panels of Figure 2 with added the fused profile and the residual error obtained with the modified**
10   **algorithm (magenta lines).**





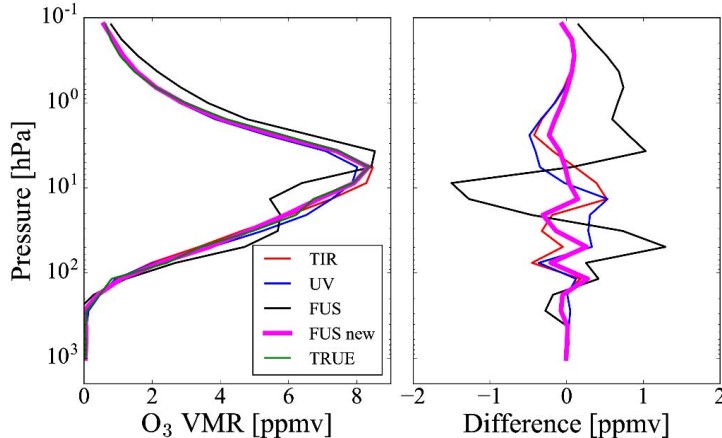

**Figure 7. Central and right panels of Figure 3 with added the fused profile and the residual error obtained with the modified**
10  **algorithm (magenta lines).**





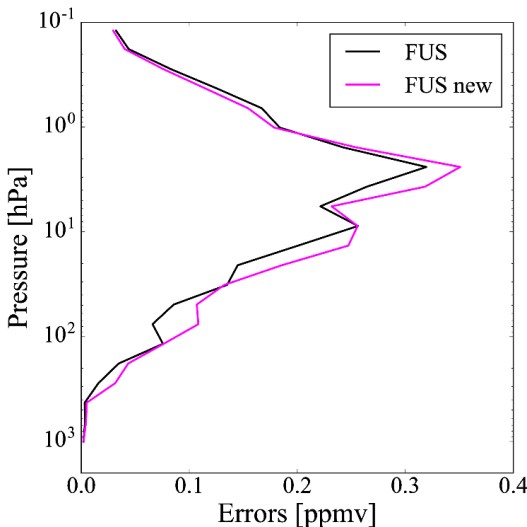

**Figure 8. Errors of the fused profile when we use the original (black line) and the generalized (magenta line) CDF for case 2.**



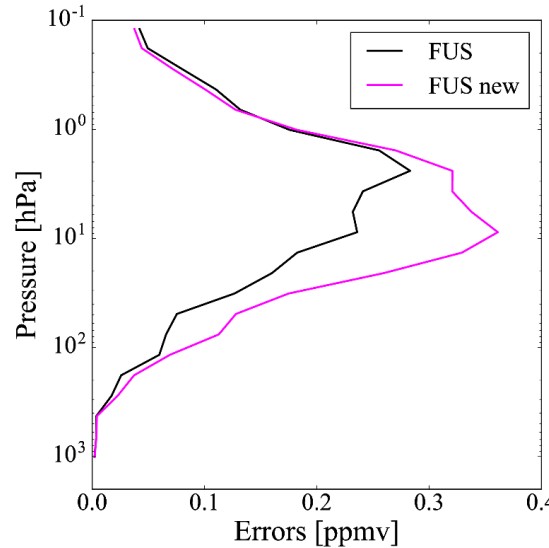

**Figure 9.** Errors of the fused profile when we use the original (black line) and the generalized (magenta line) CDF for case 3.