# Peer review of "Importance of interpolation and coincidence errors in data fusion"

_Atmospheric Measurement Techniques, 2017_

## Referee Comment (RC1) · Anonymous Referee #1 · 12 Nov 2017

Review

This is a comprehensive mathematical analysis of the data fusion methodology, with many of its features already discussed in previous papers by Ceccherini et al. The paper should be of interest to the Earth Observation community. However, before the paper is suitable for publication, the authors should address a few general points, identified immediately below, as well as the specific comments, also identified below.

As indicated in the AMTD review, the authors should address the following three points: (i) In the introduction provide more detail on how this paper is different from the previous works by Ceccherini et al. on data fusion. This should allow the reader to follow better the development of the data fusion ideas. (ii) In the discussion of previous work in the introduction, identify whether the papers discussed consider real or simulated data

[Figure]

(e.g., Natraj et al. consider simulated data). (iii) On the issue of studies using multi-spectral simulated observations (discussed in the introduction) I suggest the authors consider including the following references, which include other combinations besides IR and UV, and two review papers:

Timmermans, R., W.A. Lahoz, J.-L. Attié, V.-H. Peuch, L. Curier, D. Edwards, H. Eskes, and P. Builtjes, 2015: Observing System Simulation Experiments for Air Quality. Atmos. Env., 115, 199-213, doi:10.1016/j.atmosenv.2015.05.032. This is a review paper.

Hache, E., J.-L.Attié, C. Tournier, P. Ricaud, L. Coret, W.A. Lahoz, L. El Amraoui, B. Josse, P. Hamer, J. Warner, X. Liu, K. Chance, M. Hoepfner, R. Spurr, V. Natraj, S. Kuwalik, and A. Eldering, 2014: The added value of a geostationary thermal infrared and visible instrument to monitor ozone for air quality. Atmos. Meas. Tech., 7, 2185-2201. This paper uses the thermal infrared and the visible. Note that Natraj et al. considers other combinations besides IR and UV.

Lahoz, W.A., V.-H. Peuch, J. Orphal, J.-L. Attié, K. Chance, X. Liu, D. Edwards, H. Elbern, J.-M. Flaud, M. Claeyman, and L. El Amraoui, 2012: Monitoring air quality from space: The case for the geostationary platform. Bull. Amer. Meteorol. Soc., 93, 221-233, doi: 10.1175/BAMS-D-11-00045.1, 221-233. This is a review paper.

Specific comments:

P. 2

L. 40: Why choose 6 km for the correlation length?

P. 6

L. 19: Why choose a 5% a priori error?

P. 7

L. 23: I suggest you remove "indeed". Same elsewhere. Please omit needless words.

L. 24: Please avoid anthropomorphising the data fusion algorithm. I suggest you use a word other than "misled".

P. 17

Fig. 6 caption: "...with the fused profile added...". Same for Fig. 7 caption.
* * *

---

## Referee Comment (RC2) · Anonymous Referee #2 · 21 Nov 2017

Overview

Importance of interpolation and coincidence errors in data fusion Simone Ceccherini, Bruno Carli, Cecilia Tirelli, Nicola Zoppetti, Samuele Del Bianco, Ugo Cortesi, Jukka Kujanpää, Rossana Dragani

This is a well-written paper on a subject of interest to the atmospheric measurement community. The use of data from different instruments and different spectral regions to better constrain vertical profiles is an important step forward in improving and simplifying the production and use of data from the advanced satellite systems currently under development. The methodology appears sound and is based on an earlier development published by the authors. It is recommended that the paper be published. Some copy editing will improve the readability of the paper and detailed comments are included later.

It is not clear to this reviewer that the methodology presented actually achieves the complete goal represented by the authors. Since the two profiles being combined have been determined by constraints independently applied to the observations of the two instruments, there is no direct component of the fitting cost function that requires the simultaneous best fit to the actual observations. Of course, it is difficult to evaluate this distinction because the peculiarities of each data set (UV and TIR) are determined by the two instruments and may be difficult or impossible to model adequately in the test data sets. The recommendation here is to discuss this issue so that the good work being presented is presented in context.

Some more discussion of the appropriate interpolation approach for the covariance matrices should also be considered. The outcome of the fusion process will depend on the assumptions made regarding this process.

General comments

1. Since CDF refers to 'Complete Data Fusion' most cases referencing 'CDF' should probably be replaced by 'CDF method'.
2. This reviewer prefers to see scientific literature written in the third person unless it is absolutely necessary to do otherwise.
3. It would appear that the increase in DOFs achieved by the method is primarily due to extending the vertical range of observational response over which the system is sensitive to changes in the profiles. This might be a useful remark to include in the conclusions.

Specific comments.

1. Line 32. Missing commas in the reference call-outs.
2. Line 33. Delete 'as'

3. Line 1. '… is reduced to about one quarter with this approach.

4. Line 8. '… the advantages in using … observing ozone profiles … space exploits the synergy of measuring …'
5. Line 25. '… the TIR …'
6. Line 30.  It isn't clear to this reviewer what the reason for merging two data sets that are known to be different.
7. Line 36.  'Ceccherini et al. (2014).'
8. Line 22. '… independent, simultaneous …'
9. Line 24.  Since the estimates all contain some portion of a priori information, they are not truely independent.  Perhaps some other description would be more appropriate.  Furthermore, it is not obvious to this reviewer how to use any numerical estimate of the interdependence to improve the subsequent analysis.
10. Line 29.  '… obtained by propagating …'
11. Line 32.  Suggest: 'The CDF solutions of the profiles considered are given …'

12. Line 6. '… obtained by propagating …'
13. Line 8.  '… profiles, x, …'
14. Line 20.  '… in the presence …'

15. Line 9.  Lost comma.
16. Line 11.  Delete last equal sign.

17. Line 20.  '… in the central and right …'
18. Line 25  '… wide range …'

19. Line 33.  '… provides, in general, …'

20.  Line 9.  Suggest: 'RD contributed extensive revisions to the manuscript.'

---

## Author Comment (AC1) · 8 Jan 2018

We thank the reviewer for the useful comments. In the following, we answer the specific comments (included in "**boldface**" for clarity) and, whenever required, we describe the related changes implemented in the revised manuscript. Page and line numbers indicated refer to the original version of the paper published on AMTD.

**Anonymous Referee #1**

**Review**

**This is a comprehensive mathematical analysis of the data fusion methodology, with many of its features already discussed in previous papers by Ceccherini et al. The paper should be of interest to the Earth Observation community. However, before the paper is suitable for publication, the authors should address a few general points, identified immediately below, as well as the specific comments, also identified below.**

**As indicated in the AMTD review, the authors should address the following three points:**

**(i) In the introduction provide more detail on how this paper is different from the previous works by Ceccherini et al. on data fusion. This should allow the reader to follow better the development of the data fusion ideas.**

The new achievement presented in this paper is the solution to the problems that occur to the CDF method when the fusing profiles are either retrieved on different vertical grids or referred to different true profiles. The solution is to take into account the interpolation and the coincidence errors in the fusion method. We determine the expressions of the interpolation and coincidence errors and show how they enter in the CDF formula.
We added these details in the introduction of the revised paper (P. 2 L. 5).

**(ii) In the discussion of previous work in the introduction, identify whether the papers discussed consider real or simulated data (e.g., Natraj et al. consider simulated data).**

We included this detail in the revised paper.

**(iii) On the issue of studies using multispectral simulated observations (discussed in the introduction) I suggest the authors consider including the following references, which include other combinations besides IR and UV, and two review papers:**

**Timmermans, R., W.A. Lahoz, J.-L. Attié, V.-H. Peuch, L. Curier, D. Edwards, H. Eskes, and P. Builtjes, 2015: Observing System Simulation Experiments for Air Quality. Atmos. Env., 115, 199-213, doi:10.1016/j.atmosenv.2015.05.032. This is a review paper.**

**Hache, E., J.-L.Attié, C. Tournier, P. Ricaud, L. Coret, W.A. Lahoz, L. El Amraoui, B. Josse, P. Hamer, J. Warner, X. Liu, K. Chance, M. Hoepfner, R. Spurr, V. Natraj, S. Kuwalik, and A. Eldering, 2014: The added value of a geostationary thermal infrared and visible instrument to monitor ozone for air quality. Atmos. Meas. Tech., 7, 2185- 2201. This paper uses the thermal infrared and the visible. Note that Natraj et al. considers other combinations besides IR and UV.**

**Lahoz, W.A., V.-H. Peuch, J. Orphal, J.-L. Attié, K. Chance, X. Liu, D. Edwards, H. Elbern, J.-M. Flaud, M. Claeyman, and L. El Amraoui, 2012: Monitoring air quality from space: The case for the geostationary platform. Bull. Amer. Meteorol. Soc., 93, 221-233, doi: 10.1175/BAMS-D-11-00045.1, 221-233. This is a review paper.**

We included these references in the revised paper.

**Specific comments:**

**P. 2**
**L. 40: Why choose 6 km for the correlation length?**

The correlation length is used to reduce oscillations in the retrieved profile. A value of 6 km is typically used for nadir ozone profile retrieval.
References:
Liu, X., Bhartia, P. K., Chance, K., Spurr, R. J. D., and Kurosu, T. P.: Ozone profile retrievals from the Ozone Monitoring Instrument, Atmos. Chem. Phys., 10, 2521-2537, https://doi.org/10.5194/acp-10-2521-2010, 2010.
Kroon, M., de Haan, J. F., Veefkind, J. P., Froidevaux, L., Wang, R., Kivi, R. and Hakkarainen, J. J.: Validation of operational ozone profiles from the Ozone Monitoring Instrument, J. Geophys. Res., 116, D18305, doi:10.1029/2010JD015100, 2011.
Miles, G. M., Siddans, R., Kerridge, B. J., Latter, B. G., and Richards, N. A. D.: Tropospheric ozone and ozone profiles retrieved from GOME-2 and their validation, Atmos. Meas. Tech., 8, 385-398, https://doi.org/10.5194/amt-8-385-2015, 2015.

We added a sentence and these references to the revised paper.

**P. 6**
**L. 19: Why choose a 5% a priori error?**

For the estimation of the coincidence error we consider 5% of the a priori profile because this value is consistent with the difference between the true profiles of TIR and UV measurements used in the simulations.
We specified this in the revised paper.

**P. 7**
**L. 23: I suggest you remove "indeed". Same elsewhere. Please omit needless words.**

In the revised paper we replaced "Indeed" with "In the latter case" and we removed "if these identities are not present" at P. 7 L. 21-22.
We removed "indeed" at P. 7 L. 35.

**L. 24: Please avoid anthropomorphising the data fusion algorithm. I suggest you use a word other than "misled".**

In the revised paper a different sentence is used.

**P. 17**
**Fig. 6 caption: ": : :with the fused profile added: : :". Same for Fig. 7 caption.**

In the revised paper we rephrased captions 6 and 7.

---

## Author Comment (AC2) · 8 Jan 2018

We thank the reviewer for the useful comments. In the following, we answer the specific comments (included in "**boldface**" for clarity) and, whenever required, we describe the related changes implemented in the revised manuscript. Page and line numbers indicated refer to the original version of the paper published on AMTD.

**Anonymous Referee #2**

**Overview**

**Importance of interpolation and coincidence errors in data fusion Simone Ceccherini, Bruno Carli, Cecilia Tirelli, Nicola Zoppetti, Samuele Del Bianco, Ugo Cortesi, Jukka Kujanpää, Rossana Dragani**

**This is a well-written paper on a subject of interest to the atmospheric measurement community. The use of data from different instruments and different spectral regions to better constrain vertical profiles is an important step forward in improving and simplifying the production and use of data from the advanced satellite systems currently under development. The methodology appears sound and is based on an earlier development published by the authors. It is recommended that the paper be published. Some copy editing will improve the readability of the paper and detailed comments are included later.**

**It is not clear to this reviewer that the methodology presented actually achieves the complete goal represented by the authors. Since the two profiles being combined have been determined by constraints independently applied to the observations of the two instruments, there is no direct component of the fitting cost function that requires the simultaneous best fit to the actual observations. Of course, it is difficult to evaluate this distinction because the peculiarities of each data set (UV and TIR) are determined by the two instruments and may be difficult or impossible to model adequately in the test data sets. The recommendation here is to discuss this issue so that the good work being presented is presented in context.**

We understand that the reviewer is asking whether the proposed data fusion method is achieving the goal of obtaining results as good as those of the simultaneous best fit (simultaneous retrieval). The equivalence between the CDF method and the simultaneous retrieval has been discussed in Ceccherini et al. (2015). In that paper the two methods were analytically proved to be equivalent if a linear approximation can be applied to the forward models. The equivalence was also verified with a real measurement of the MIPAS instrument. The goal of this new paper is the solution of the problems that occur when the fusing profiles are either retrieved on different vertical grids or referred to different true profiles. The solution is to take into account the interpolation and the coincidence errors in the fusion method.
As also suggested by reviewer #1 we have better specified the goal of this paper in the introduction of the revised paper (Page 2 Line 5).

**Some more discussion of the appropriate interpolation approach for the covariance matrices should also be considered. The outcome of the fusion process will depend on the assumptions made regarding this process.**

As we can see from Eq. (8) the only quantities that need to be interpolated in case of different vertical grids are the averaging kernel matrices, therefore, no interpolation approach is applied to the covariance matrices.

We specified this in the revised paper (Page 4 Line 13).

**General comments**

**1. Since CDF refers to 'Complete Data Fusion' most cases referencing 'CDF' should probably be replaced by 'CDF method'.**

We replaced "CDF" with "CDF method" in most cases in the revised paper.

**2. This reviewer prefers to see scientific literature written in the third person unless it is absolutely necessary to do otherwise.**

In a few cases, the sentences were changed in the revised paper in order to remove "we". However, the first person is largely used in this journal and the change was not done in all sentences.

**3. It would appear that the increase in DOFs achieved by the method is primarily due to extending the vertical range of observational response over which the system is sensitive to changes in the profiles. This might be a useful remark to include in the conclusions.**

In order to see how the DOF are distributed in altitude, in Fig. A we report the diagonal elements of the averaging kernel matrices of the ozone profiles obtained from the TIR measurement, from the UV measurement and from the data fusion in the case 1 (same true profile and same vertical grid).

[Figure]

Fig. A. Diagonal elements of the averaging kernel matrices of the ozone profiles obtained from the TIR measurement (red line), from the UV measurement (blue line) and from the data fusion (black line) in the case 1.

We can see that the increase in DOF obtained by the fusion is distributed along all the vertical range. An extension of the vertical range is observed with respect to the UV measurement, but it does not seem to be the main effect.

**Specific comments.**

**Page 1**

**1. Line 32. Missing commas in the reference call-outs.**

We made the correction in the revised paper.

**2. Line 33. Delete 'as'**

The first "as" has been replaced with "since" in the revised paper.

**Page 2**

**3. Line 1. '… is reduced to about one quarter with this approach.**

We replaced with "…is reduced of about a quarter with this approach."

**4. Line 8. '… the advantages in using … observing ozone profiles … space exploits the synergy of measuring …'**

We modified this sentence taking into account also a comment of reviewer #1.

**5. Line 25. '… the TIR …'**

We prefer to leave it as it is.

**6. Line 30. It isn't clear to this reviewer what the reason for merging two data sets that are known to be different.**

It is not frequent to have different measurements with exactly the same geolocation. Therefore, the measurements that are fused are generally selected applying a coincidence criterion, which requires reciprocal space-time distances to be less than some defined thresholds. In this case the geolocations are different, even if within an acceptable range, and, consequently, the related measurements can correspond to different true profiles.
We added a sentence in the introduction of the revised paper (Page 2 Line 5).

**7. Line 36. 'Ceccherini et al. (2014).'**

We made the correction in the revised paper.

**Page 3**

**8. Line 22. '… independent, simultaneous …'**

We replaced with "… independent and simultaneous …" in the revised paper.

**9. Line 24. Since the estimates all contain some portion of a priori information, they are not truly independent. Perhaps some other description would be more appropriate. Furthermore, it is not obvious to this reviewer how to use any numerical estimate of the interdependence to improve the subsequent analysis.**

The reviewer is right, the estimates are not independent when the same a priori information is used in the single retrievals. However, since the CDF method removes the a priori information used in the single retrievals a numerical estimate of the interdependence is not needed: that is, even if the

retrieved profiles are not independent because of the common "a-priori" (our choice, but not a necessary choice), the fused quantities (Eq. (4)) are independent quantities.

We removed "independent" in the revised paper.

Page 4 line 4: a sentence was modified to underline the independence of the fused quantities defined in Eq. (4).

**10. Line 29. '… obtained by propagating …'**

We made the correction in the revised paper.

**11. Line 32. Suggest: 'The CDF solutions of the profiles considered are given …'**

In the revised paper we modified the sentence in "The CDF solution for the considered profiles is given …".

**Page 4**

**12. Line 6. '… obtained by propagating …'**

We made the correction in the revised paper.

**13. Line 8. '… profiles, x, …'**

We think that the commas are not needed.

**14. Line 20. '… in the presence …'**

We made the correction in the revised paper.

**Page 5**

**15. Line 9. Lost comma.**

We think that no comma is lost here.

**16. Line 11. Delete last equal sign.**

We prefer to leave it.

**Page 6**

**17. Line 20. '… in the central and right …'**

We made the correction in the revised paper.

**18. Line 25 '… wide range …'**

We made the correction in the revised paper.

**Page 7**

**19. Line 33. '… provides, in general, …'**

In the revised paper we modified the sentence in "…the fusion generally provides …"

**Page 8**

**20. Line 9. Suggest: 'RD contributed extensive revisions to the manuscript.'**

We prefer to leave it as it is.